

# Spatial and temporal dynamics of SAR11 marine bacteria across a nearshore to offshore transect in the tropical Pacific Ocean

Sarah J. Tucker[1,2], Kelle C. Freel[1], Elizabeth A. Monaghan[1,2], Clarisse E. S. Sullivan[1,3], Oscar Ramfelt[1,3], Yoshimi M. Rii[1,4] and Michael S. Rappé[1]

[1] Hawai'i Institute of Marine Biology, School of Ocean and Earth Science and Technology, University of Hawai'i at Mānoa, Kāne'ohe, Hawai'i, United States
[2] Marine Biology Graduate Program, University of Hawai'i at Mānoa, Honolulu, Hawai'i, United States
[3] Department of Oceanography, School of Ocean and Earth Science and Technology, University of Hawai'i at Mānoa, Honolulu, Hawai'i, United States
[4] He'eia National Estuarine Research Reserve, Kāne'ohe, Hawai'i, United States

Corresponding author
Michael S. Rappé, rappe@hawaii.edu

## ABSTRACT

Surveys of microbial communities across transitions coupled with contextual measures of the environment provide a useful approach to dissect the factors determining distributions of microorganisms across ecological niches. Here, monthly time-series samples of surface seawater along a transect spanning the nearshore coastal environment within Kāne'ohe Bay on the island of O'ahu, Hawai'i, and the adjacent offshore environment were collected to investigate the diversity and abundance of SAR11 marine bacteria (order Pelagibacterales) over a 2-year time period. Using 16S ribosomal RNA gene amplicon sequencing, the spatiotemporal distributions of major SAR11 subclades and exact amplicon sequence variants (ASVs) were evaluated. Seven of eight SAR11 subclades detected in this study showed distinct subclade distributions across the coastal to offshore environments. The SAR11 community was dominated by seven (of 106 total) SAR11 ASVs that made up an average of 77% of total SAR11. These seven ASVs spanned five different SAR11 subclades (Ia, Ib, IIa, IV, and Va), and were recovered from all samples collected from either the coastal environment, the offshore, or both. SAR11 ASVs were more often restricted spatially to coastal or offshore environments (64 of 106 ASVs) than they were shared among coastal, transition, and offshore environments (39 of 106 ASVs). Overall, offshore SAR11 communities contained a higher diversity of SAR11 ASVs than their nearshore counterparts, with the highest diversity within the little-studied subclade IIa. This study reveals ecological differentiation of SAR11 marine bacteria across a short physiochemical gradient, further increasing our understanding of how SAR11 genetic diversity partitions into distinct ecological units.

## INTRODUCTION

The SAR11 order Pelagibacterales of the class Alphaproteobacteria is one of the most abundant and ubiquitous bacterial lineages on Earth (*Morris et al., 2002*). While found throughout the global ocean and freshwater environments, SAR11 bacteria are particularly abundant in stratified, oligotrophic surface oceans, often making up 25% or more of all bacterioplankton cells (*Morris et al., 2002*). SAR11 are chemoheterotrophic, free-living microorganisms that are uniquely adapted to nutrient-poor environments through small cell sizes and streamlined genomes (*Rappé et al., 2002*; *Grote et al., 2012*; *Noell & Giovannoni, 2019*). Similar to other abundant marine bacteria and archaea, most SAR11 lineages are difficult to culture in the laboratory (*Swan et al., 2013*), and the current capacity to interrogate SAR11 strains in a controlled laboratory setting remains limited to a small portion of the total SAR11 phylogenetic breadth (*e.g.*, *Jimenez-Infante et al., 2017*; *Henson et al., 2018*; *Monaghan, Freel & Rappé, 2020*).

The SAR11 clade is genetically diverse, with up to 18% small subunit (SSU) rRNA gene sequence divergence distributed among at least five major subgroups (*Brown et al., 2012*) and 10 subclades including Ia, Ib, Ic, IIa, IIb, IIIa, IIIb, IV, Va, and Vb (*Giovannoni, 2017*). It remains uncertain, however, if all five major subgroups derive from a single monophyletic lineage within the Alphaproteobacteria as compositional biases and long evolutionary branches have made it a difficult phylogeny to resolve, particularly with respect to subgroup V (*Muñoz-Gómez et al., 2019*). Some SAR11 subclades exhibit spatiotemporal distributions that are delineated by depth, season, or geographical location (*Field et al., 1997*; *Carlson et al., 2008*; *Brown et al., 2012*), revealing ecological differentiation at a broad phylogenetic level. The ecotype concept describes an ecologically homogeneous group of closely related bacteria whose genetic diversity is guided by cohesive forces such as periodic selection, recombination, and genetic drift (*Cohan, 2006*; *Koeppel et al., 2008*). This concept has been helpful in discerning populations among highly diverse and widely-distributed bacterial groups such as *Prochlorococcus* (*Biller et al., 2014*) and *Bacillus* (*Kopac et al., 2014*). Previous studies of SAR11 have shown evidence for ecotypic differentiation (*Brown et al., 2012*; *Vergin et al., 2013*; *Tsementzi et al., 2019*; *Kraemer et al., 2019*). Yet, other research has attributed at least a portion of the genetic diversity harbored by the SAR11 lineage to neutral processes (*Hellweger, van Sebille & Fredrick, 2014*; *Manrique & Jones, 2017*).

The genetic diversity within SAR11 has presented a challenge in understanding the ecological roles and evolutionary origins of the lineage. For example, rather than gene content, measures of SAR11 genomic microdiversity including single-nucleotide polymorphisms and single-amino acid variants were necessary to characterize two ecological niches among closely related populations within the global Ia.3.V subgroup (*Delmont et al., 2019*). High intra-population genomic variation has made it difficult to reconstruct SAR11 genomes from metagenomic data (*Tully, Graham & Heidelberg, 2018*; *Delmont et al., 2019*). Even when SAR11 genomes have been reconstructed from environmental samples, understanding the boundaries between the sympatric populations that they represent has proven challenging (*Delmont et al., 2019*).

16S rRNA gene sequencing surveys have been central to defining SAR11 subclades and examining their spatiotemporal distributions in the environment (*e.g.*, *Field et al., 1997*; *Carlson et al., 2008*; *Brown et al., 2012*; *Vergin et al., 2013*; *Salter et al., 2014*; *Herlemann et al., 2014*; *West et al., 2016*). For example, a multi-year study from the Bermuda Atlantic Time-Series (BATS) in the Sargasso Sea has revealed that the relative abundance of some SAR11 subclades changes with depth and seasonal regimes (*Carlson et al., 2008*; *Vergin et al., 2013*). In other work, a combination of 16S rRNA gene and internal genomic spacer (ITS) analyses were able to discern cold-water (Ia.1) and warm-water (Ia.3) variants of SAR11 subclade Ia (*Brown et al., 2012*). Despite continued efforts to investigate SAR11 bacteria through 16S rRNA gene and ITS surveys (*Brown & Fuhrman, 2005*; *Ngugi & Stingl, 2012*; *Needham, Sachdeva & Fuhrman, 2017*), fluorescence *in situ* hybridization (*Alonso-Sáez et al., 2007*; *Salcher, Pernthaler & Posch, 2011*), the cultivation of new strains (*Jimenez-Infante et al., 2017*; *Henson et al., 2018*; *Monaghan, Freel & Rappé, 2020*), metagenomics (*Tsementzi et al., 2016*; *Tsementzi et al., 2019*; *Delmont et al., 2019*; *Haro-Moreno et al., 2019*), and single-cell sequencing efforts (*Thrash et al., 2014*; *Thompson et al., 2019*; *Pachiadaki et al., 2019*), the spatiotemporal characterization of the majority of SAR11 diversity, particularly outside of subclade Ia, remains limited.

One useful approach to investigate the interplay between ecology and evolution within natural microbial communities is to sample across environmental gradients. In the marine environment, light, temperature, pressure, nutrients and other variables form steep gradients with depth in the water column and have been used to investigate the partitioning of microorganisms within different depth horizons of stratified offshore systems (*Morris et al., 2005*; *Cram et al., 2015*; *Haro-Moreno et al., 2018*; *Mende, Boeuf & DeLong, 2019*). The nearshore to offshore transition can also offer steep physiochemical gradients in nutrients, salinity, biomass, and productivity which are likely to impact the structure of microbial communities (*Herlemann et al., 2014*; *Wang et al., 2019b*). In tropical coastal environments like those found on islands and atolls, this gradient can be acute; near-island biological, anthropogenic, and physical oceanographic processes provide a substantial source of nutrients for increased biological productivity in otherwise oligotrophic oceanic waters (*Gove et al., 2016*).

Kāneʻohe Bay is a well-studied, semi-enclosed embayment on the windward side of the island of Oʻahu, Hawaiʻi (*Bahr, Jokiel & Toonen, 2015*). In this study, we used the steep physiochemical gradient provided by the transition from the nearshore environment within Kāneʻohe Bay to the adjacent offshore environment as a natural laboratory to investigate SAR11 marine bacteria across space and time. This system was sampled monthly over 2 years to capture the microbial community from 10 sites spanning the interior of the bay and the surrounding offshore. We used 16S rRNA gene amplicon sequencing to assess the distribution of SAR11 across a phylogenetic resolution that spanned subclades to individual SAR11 amplicon sequence variants (ASVs). Our findings provide new insight into the ecological differentiation of SAR11 subclades and the distribution of these subclades across the nearshore to offshore continuum.

## MATERIALS AND METHODS

### Sample collection and environmental parameters

Between August 2017 and June 2019, seawater was collected from a depth of 2 m at 10 sites in and around Kāneʻohe Bay, Oʻahu, Hawaiʻi, on a near-monthly basis (20 sampling events over 23 months; Fig. 1). At each station, seawater samples for biogeochemical analyses and nucleic acids were collected, and *in situ* measurements of seawater temperature, pH, and salinity were made with a YSI 6,600 sonde (YSI Incorporated, Yellow Springs, OH, USA). Approximately one L of seawater was prefiltered using an 85 μm Nitex mesh and subsequently collected on a 25-mm diameter 0.1-μm pore-sized polyethersulfone (PES) membrane for nucleic acids (Supor-100, Pall Gelman Inc., Ann Arbor, MI, USA). The filters were submerged in DNA lysis buffer (*Suzuki et al., 2001*; *Yeo et al., 2013*) and stored at −80 °C until extraction.

Subsamples for chlorophyll *a* were collected by filtering 125 mL of seawater onto 25-mm diameter GF/F glass microfiber filters (Whatman, GE Healthcare Life Sciences, Chicago, IL, USA), and stored in aluminum foil at −80 °C until extraction in 100% acetone and subsequent measurement of fluorescence with a Turner 10AU fluorometer (Turner Designs, Sunnyvale, CA, USA) followed standard techniques (*Welschmeyer, 1994*). Seawater for cellular enumeration was preserved in two-mL aliquots in a final concentration of 0.95% (v:v) paraformaldehyde (Electron Microscopy Services, Hatfield, PA, USA) at −80 °C until analyzed *via* flow cytometry. Cellular enumeration of cyanobacterial picophytoplankton (marine *Synechococcus* and *Prochlorococcus*), eukaryotic picophytoplankton, and non-cyanobacterial (presumably heterotrophic) bacteria and archaea (hereafter referred to as heterotrophic bacteria) was performed on an EPICS ALTRA flow cytometer (Beckman Coulter Inc., Brea, CA, USA), following the method of Monger and Landry (*Monger & Landry, 1993*).

Seasons were delineated by fitting a harmonic function to surface seawater temperature collected hourly between 2010–2019 at NOAA station MOKH1 in Kāneʻohe Bay (https://www.ndbc.noaa.gov/station_page.php?station=mokh1; Fig. S1A). Transition seasons (spring and fall) typically experience the greatest amount of change in seawater temperature. Thus, using the time-derivative of the function, thresholds were defined where the derivative was at its greatest absolute value: ≥0.0255 (spring; 30 March through 27 June 2017–19) and ≤−0.0255 (fall; 29 September through 26 December 2017–19; Fig. S1B). Summer and winter were defined as the periods between the thresholds (*i.e.*, when the derivative was <0.0255 to >−0.0255; summer: 28 June through 28 September 2017–19 and winter: 27 December through 29 March 2018–19).

In RStudio (version 1.1.456) (*R Core Team, 2018*) with the Vegan package (*Oksanen et al., 2019*), stations were grouped into environment types based on non-metric rank-based analysis using the 'metaMDS' function with k = 2 and a Bray–Curtis transformed distance matrix of environmental parameters (surface seawater temperature, pH, salinity), chlorophyll *a* concentrations, and cellular abundances of *Prochlorococcus, Synechococcus*, eukaryotic picophytoplankton, and heterotrophic bacteria. The pairwiseAdonis package (*Martinez Arbizu, 2020*) and the 'betdisper' function
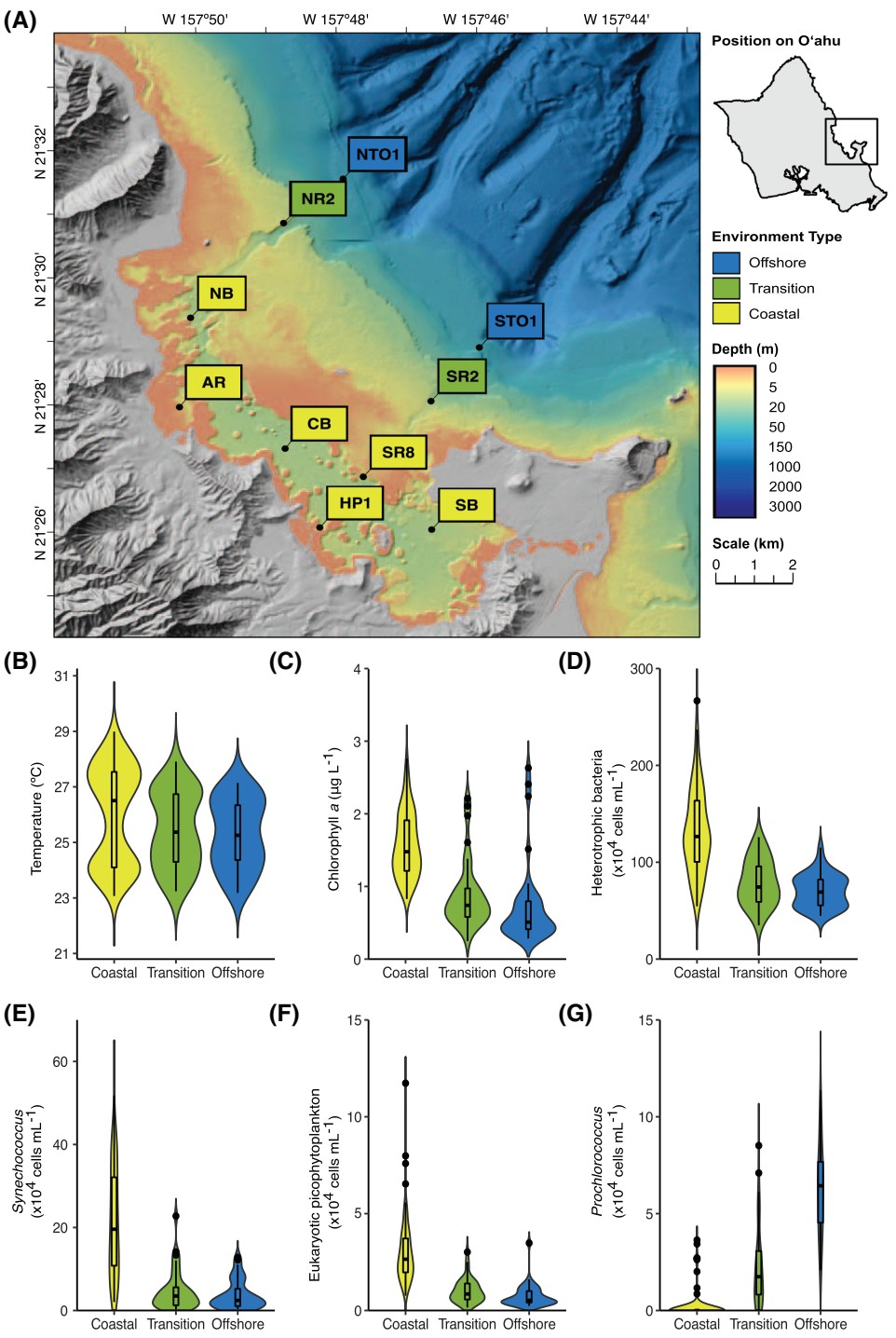

**Figure 1 Location and characteristics of the Kāne'ohe Bay Time-series (KByT) sampling stations.** (A) Map of sampling stations within and immediately adjacent to Kāne'ohe Bay on the island of O'ahu, Hawai'i. (B–G) Distribution of environmental parameters over 2 years of sampling across the coastal, transition, and offshore regions of KByT, including (B) temperature, (C) chlorophyll *a*, (D) cellular abundance of heterotrophic bacteria, (E) cellular abundance of *Synechococcus*, (F) cellular abundance of eukaryotic picophytoplankton, and (G) cellular abundance of *Prochlorococcus*. Box plots show the mean and first and third quartile. Map plotted in ArcGIS Pro 2.7.

in the Vegan package were used to evaluate the statistical significance and dispersion of these groupings. Comparisons of environmental variables, cellular abundances, and sequencing depth across the spatial and temporal groupings were conducted using the package multcomp (*Hothorn, Bretz & Westfall, 2008*) with one-way ANOVAs testing for multiple comparisons of means with Holm correction and Tukey contrasts.

## DNA extraction and sequencing

Genomic DNA was extracted using a Qiagen Blood and Tissue Kit (Qiagen Inc., Valencia, CA, USA) with modifications (*Becker, Brandon & Rappé, 2007*). For each sample, 16S rRNA gene fragments were amplified by polymerase chain reaction using a dual-index sequencing strategy where barcoded universal primers 515-Y-F and 926R (*Parada, Needham & Fuhrman, 2016*) are complete with Illumina sequencing adapters, barcode, and index. The 25 µL reactions included 13 µL $H_2O$, 0.5 µL each of forward and reverse primer at 0.2 µm final concentration, one µL gDNA (0.5 ng), and 10 µL 1 × 5PRIME Hot Master Mix (0.5 U *Taq* DNA polymerase, 45 mm KCl, 2.5 mm $Mg^{2+}$, 200 µm dNTPs) (Quantabio, Beverly, MA, USA). PCR conditions included an initial denaturation at 95 °C for 2 min followed by 30 cycles of 95 °C for 45 s, 50 °C for 45 s, and 68 °C for 90 s, and a final 5 min extension at 68 °C. PCR products were inspected on a 1.5% agarose gel and quantified using the Qubit dsDNA HS Kit (Qubit 2.0, Life Technologies, Foster City, CA, USA). PCR products were normalized to 240 ng each, pooled, and purified using the QIAquick PCR Purification Kit (Qiagen, Hilden, Germany). Pooled libraries were then sequenced on an Illumina MiSeq v2 250 bp paired-end run at the Oregon State University Center for Genome Research & Biocomputing.

## Sequence analysis

The sequence data were demultiplexed and assessed for quality in Qiime2 v 2019.4 (*Bolyen et al., 2019*). Forward reads were truncated to 245 bp using the–p-trunc-len command in Qiime2 to improve their quality. Reverse reads were not used in these analyses due to poor quality. Sequences were denoised using DADA2 (*Callahan et al., 2016*) and delineated into exact amplicon sequence variants (ASVs). Denoised sequence data were then assigned taxonomy using SILVA v132 as a reference database (*Quast et al., 2012*). Mock communities were assessed to establish that the denoising of sequences using single-end reads was representative of taxa and abundances expected in mock community samples.

ASVs that contained a minimum of 10 reads in at least two independent samples were retained for subsequent analyses. ASVs undefined at the level of domain and uncharacterized at the level of phylum were excluded from subsequent analyses. Sequences that matched to chloroplasts or mitochondria were included in analyses of sequencing depth (Tables S2–S4), ASV occurrence (Table S5), and community structure (Figs. 2A, S4B, Table S6), but removed in all subsequent analyses of SAR11 including calculations of the relative proportion of SAR11 within the microbial community and DESeq2 analysis of differences in SAR11, *Prochlorococcus*, and *Synechococcus* abundance across environments and seasons.

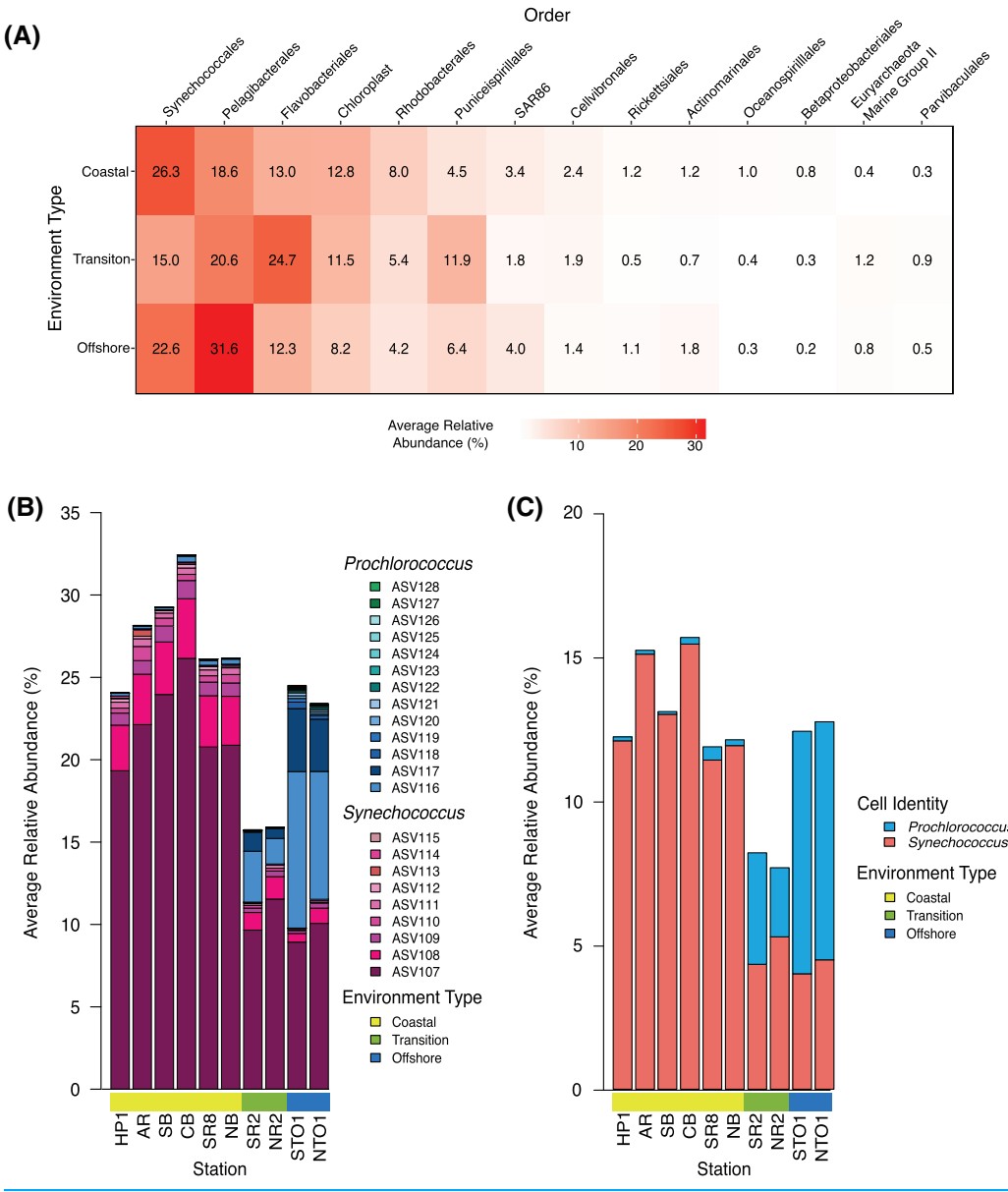

**Figure 2  Community structure of dominant groups across environment types in KByT.** (A) Average relative abundance of the 14 most abundant order-level 16S rRNA gene-based groups across coastal (*n* = 120), transition (*n* = 40), and offshore (*n* = 40) environments sampled over 2 years of KByT. Taxonomy based on Silva v132. (B) Distribution of *Synechococcus* and *Prochlorococcus* ASVs across KByT. *n* = 20 samples per station. (C) Distribution of *Synechococcus* and *Prochlorococus* cellular abundances relative to the total prokaryotic community. *n* = 20 samples per station.

ASVs classified by the SILVA v132 database as the bacterial order SAR11 and the family AEGEAN-169 within the order Rhodospirillales (for SAR11 subgroup V) were assigned to SAR11 subclades using phylogenetic placement methods. First, a reference alignment was created in ARB (*Ludwig, 2004*) from 16S rRNA gene sequences downloaded from NCBI and SAR11 ASVs identified in this study. Reference sequences were selected based on similarity to ASVs, and also included all known cultivated SAR11 strains.

Two outgroup representatives were included: *Escherichia coli* (NR_024570.1) and *Magnetococcus marinus* (NR_074371.1). We applied a mask to the alignment using filter by frequency in ARB in order to remove columns containing gaps or no data, and exported the resulting 1,224 bp alignment. The mask did not remove any columns in the 245 bp region where the ASVs aligned. Using only the aligned reference sequences, the best reference tree and model parameters were selected by performing tree inference with a random starting tree, maximum-likelihood estimate of substitution rates and nucleotide frequencies, a general time-reversible + gamma model, and 100 non-parametric bootstrap replicates *via* RAxML-NG (*Kozlov et al., 2019*).

Next, the SAR11 ASVs aligned to the reference sequences were phylogenetically placed using the EPA-NG algorithm (*Barbera et al., 2018*), specifying the model parameters (GTR (0.960433/4.369000/2.298538/0.794304/6.601978/1.000000) + FU (0.271858/0.192063/0.300579/0.235499) + G4m (0.276190)), which were previously determined when conducting tree inference with RAxML-NG. Placement results of the ASVs on the reference tree were visualized in GAPPA (*Czech & Stamatakis, 2019*) using the function 'gappa examine graft' (Fig. S2) and likelihood weights of each ASV placement to SAR11 subclades were evaluated using 'gappa examine assign'. The tree was manipulated in FigTree v1.4.3. (available from http://tree.bio.ed.ac.uk/software/figtree/) to improve visualization.

Statistical analyses and visualizations were conducted in R packages phyloseq (*McMurdie & Holmes, 2013*), plotly (*Sievert, 2013*), ggplot2 (*Wickham, 2016*), and pheatmap (*Kolde, 2019*), as well as the online tool Venny 2.1 (*Oliveros, 2017*). Generalized linear models (GLMs) were built in mvabund (*Wang et al., 2019a*) using the 'manyglm' function to test for differences in the number of ASVs belonging to SAR11 subclades across seasons and environments. DESeq2 (*Love, Huber & Anders, 2014*) was used to normalize ASV abundance and to evaluate whether individual ASVs exhibited significantly different distributions across seasons or environments using Wald Tests, Bonferroni correction for multiple comparisons, and a corrected $p$-value of 0.05. The same approach was used to normalize abundance and test for spatiotemporal differences in the top 14 most abundant bacterial orders, SAR11 subclades, and total SAR11, *Synechococcus*, *Prochlorococcus*, and chloroplast abundance.

The frequency of SAR11 ASVs was assessed by segregating the ASVs into rare (0–5%), low-frequency (>5–25%), mid-frequency (25–75%), and high-frequency (75–100%) categories. Frequency was calculated as the number of samples an ASV was detected in per environment, divided by the number of total samples for a given environment (coastal $n = 120$; transition $n = 40$; offshore $n = 40$), multiplied by 100.

Variance stabilized transformed DESeq2 normalized counts of SAR11 subclades and dominant ASVs and picocyanobacteria along with environmental parameters were used to conduct Spearman's Rank correlation analysis using the 'cor' function in the stats package (*R Core Team, 2018*). Using a matrix of the Spearman's correlation coefficients, between-group average (UPGMA) linkage hierarchical clustering was conducted using the function 'hclust' in the stats package (*R Core Team, 2018*). This dendrogram was then used to order the correlation coefficients matrix for visualization.

SAR11 ASVs were compared to SAR11 isolates by aligning the SAR11 ASVs to all publicly available 16S rRNA gene sequences from SAR11 isolates using the MUSCLE algorithm in MEGAX (*Kumar et al., 2018*).

## Data availability

Amplicon sequencing data are available in the National Center for Biotechnology Information (NCBI) Sequencing Read Archive (SRA) under BioProject number PRJNA706753.

## RESULTS

### Environmental conditions

Non-DNA sequence-based parameters partitioned the Kāneʻohe Bay Time-series (KByT) sampling sites into three spatial groups, hereafter referred to as coastal (to define the nearshore grouping), transition, and offshore (Fig. S3). A pairwise-adonis analysis showed statistically significant differences among the three groups (coastal *vs.* transition: $p = 0.001$, $R^2 = 0.31$; coastal *vs.* offshore: $p = 0.001$, $R^2 = 0.43$; transition *vs.* offshore: $p = 0.012$, $R^2 = 0.071$). While significant, we note that the difference between the transition and offshore sampling sites is small. Dispersion tests found non-significant dispersion ($p = 0.115$, F-value = 2.19).

Compared to the transition and offshore stations, coastal stations were defined by lower salinity, higher chlorophyll *a* concentrations, higher cellular abundances of heterotrophic bacteria, *Synechococcus,* and eukaryotic picophytoplankton, and lower cellular abundances of *Prochlorococcus* (Fig. 1, Tables 1 & S1). The cellular abundance of *Prochlorococcus* was significantly different across all three environments (one-way ANOVA with Tukey's multiple comparisons test, $p < 0.001$), ranging from $0.2 \pm 0.7 \times 10^4$ cells mL$^{-1}$ at coastal stations (mean ± standard deviation (s.d.), $n = 120$) to $2.2 \pm 2.1 \times 10^4$ cells mL$^{-1}$ at transition stations ($n = 40$) and $6.2 \pm 2.3 \times 10^4$ cells mL$^{-1}$ at offshore stations ($n = 40$). pH was significantly different between coastal ($7.9 \pm 0.2$, $n = 120$) and offshore stations ($8.0 \pm 0.2$, $n = 40$, $p = 0.002$) and transition ($7.9 \pm 0.2$, $n = 40$) and offshore stations ($p = 0.015$); no significant difference in pH was detected between coastal and transition stations ($p = 0.841$). While not statistically significant, transition stations had lower salinity, higher chlorophyll *a* concentrations, and higher cellular abundances of heterotrophic bacteria, *Synechococcus,* and eukaryotic picophytoplankton in comparison to offshore stations (Table 1). Sea surface water temperature did not statistically differ across the three environments.

An increase in salinity and cellular abundances of heterotrophic bacteria and *Synechococcus* were observed during spring (Tables 1 & S1). At coastal stations, salinity ranged from $34.0 \pm 1.2$ (mean ± s.d., $n = 24$) during summer to $35.1 \pm 1.3$ ($n = 36$) during spring, with significant differences between spring *vs.* summer (one-way ANOVA with Tukey's multiple comparisons test, $p < 0.001$), fall ($p = 0.037$), and winter ($p = 0.002$). Salinity also reached its highest during spring at transition ($35.5 \pm 1.2$, $n = 12$) and offshore ($35.6 \pm 1.2$, $n = 12$) stations. Increases in the cellular abundance of heterotrophic bacteria between spring and fall were observed in the coastal ($p = 0.033$) and in the offshore

**Table 1 Environmental parameters and cellular abundances from KByT.**

| | Coastal | | | | Transition | | | | Offshore | | | |
|---|---|---|---|---|---|---|---|---|---|---|---|---|
| | Summer (n = 24) | Fall (n = 24) | Winter (n = 36) | Spring (n = 36) | Summer (n = 8) | Fall (n = 8) | Winter (n = 12) | Spring (n = 12) | Summer (n = 8) | Fall (n = 8) | Winter (n = 12) | Spring (n = 12) |
| Seawater temp. (°C) | 27.6 ± 0.6 | 26.5 ± 1.3 | 24.0 ± 0.4 | 26.5 ± 1.5 | 27.0 ± 0.6 | 26.3 ± 0.9 | 24.0 ± 0.5 | 25.4 ± 1.0 | 26.7 ± 0.5 | 26.2 ± 0.8 | 23.9 ± 0.6 | 25.2 ± 0.8 |
| pH | 7.9 ± 0.1 | 7.9 ± 0.2 | 8.0 ± 0.1 | 7.9 ± 0.2 | 8.0 ± 0.1 | 7.9 ± 0.3 | 8.0 ± 0.1 | 7.8 ± 0.2 | 8.1 ± 0.2 | 8.0 ± 0.2 | 8.1 ± 0.1 | 7.9 ± 0.1 |
| Salinity | 34.0 ± 1.2 | 34.4 ± 0.6 | 34.2 ± 0.6 | 35.1 ± 1.3 | 34.4 ± 0.8 | 35.2 ± 0.5 | 34.9 ± 0.5 | 35.5 ± 1.2 | 34.4 ± 0.8 | 35.3 ± 0.7 | 35.0 ± 0.4 | 35.6 ± 1.2 |
| Chlorophyll $a$ (µg L$^{-1}$) | 1.6 ± 0.4 | 1.5 ± 0.4 | 1.7 ± 0.5 | 1.5 ± 0.4 | 0.9 ± 0.5 | 0.7 ± 0.3 | 0.8 ± 0.3 | 0.9 ± 0.6 | 0.5 ± 0.3 | 0.8 ± 0.7 | 0.6 ± 0.3 | 0.8 ± 0.8 |
| Heterotrophic bacteria (×10$^6$ cells mL$^{-1}$) | 1.4 ± 0.5 | 1.1 ± 0.5 | 1.3 ± 0.4 | 1.4 ± 0.4 | 0.8 ± 0.2 | 0.7 ± 0.4 | 0.7 ± 0.2 | 0.9 ± 0.2 | 0.7 ± 0.2 | 0.6 ± 0.2 | 0.7 ± 0.2 | 0.8 ± 0.2 |
| *Synechococcus* (×10$^4$ cells mL$^{-1}$) | 26.6 ± 12.7 | 16.8 ± 13.5 | 16.5 ± 12.8 | 25.6 ± 9.7 | 4.0 ± 7.8 | 6.2 ± 7.8 | 1.8 ± 1.2 | 6.8 ± 4.5 | 4.3 ± 3.7 | 4.3 ± 3.7 | 1.6 ± 2.0 | 5.3 ± 4.0 |
| Eukaryotic picophytoplankton (×10$^4$ cells mL$^{-1}$) | 2.9 ± 1.2 | 2.8 ± 1.2 | 2.9 ± 2.4 | 3.2 ± 0.9 | 0.9 ± 0.4 | 1.3 ± 1.0 | 0.9 ± 0.6 | 1.1 ± 0.6 | 0.6 ± 0.5 | 0.7 ± 0.1 | 0.9 ± 0.9 | 0.8 ± 0.4 |
| *Prochlorococcus* (×10$^4$ cells mL$^{-1}$) | 0.1 ± 0.7 | 0.7 ± 0.1 | 0.0 ± 0.0 | 0.0 ± 0.0 | 2.8 ± 3.3 | 2.8 ± 2.4 | 1.8 ± 1.5 | 2.0 ± 1.4 | 6.9 ± 2.7 | 5.7 ± 3.1 | 6.9 ± 0.2 | 5.5 ± 1.9 |

**Note:**
Samples are averaged (mean ± s.d.) over environmental category and season.

environments ($p = 0.037$). The abundance of *Synechococcus* cells increased during spring (25.6 ± 9.7 × 10$^4$ cells mL$^{-1}$, $n = 36$) and summer (26.6 ± 12.7 × 10$^4$ cells mL$^{-1}$, $n = 24$) in the coastal environment in comparison to fall (16.8 ± 13.5 × 10$^4$ cells mL$^{-1}$, $n = 24$) and winter (16.5 ± 12 × 10$^4$ cells mL$^{-1}$, $n = 36$; $p = 0.012$). In contrast, the abundance of *Prochlorococcus* cells increased in the coastal environment during fall (0.7 ± 0.1 × 10$^4$ cells mL$^{-1}$, $n = 24$). pH, chlorophyll $a$ concentration, and the abundance of eukaryotic picophytoplankton cells showed no significant changes over seasons. Surface seawater temperatures were significantly different across all but two seasonal comparisons (spring *vs.* fall in the coastal environment ($p = 0.876$) and summer *vs.* fall in the transition and offshore environments ($p = 0.093$ and $p = 0.157$, respectively)).

## Community composition

Samples were sequenced to a depth of 25,684 ± 11,226 quality-controlled reads (mean ± s.d., $n = 200$), with a range of 9,393 to 100,278 (Table S2). Read depth did not differ across environment type or station (one-way ANOVA with Tukey's multiple comparisons test; $p = 1$ all comparisons, Table S3 & S4), although significant variation was detected among some seasonal comparisons (spring *vs.* fall, winter *vs.* fall, summer *vs.* spring, and winter *vs.* summer, $p < 0.001$; Table S3). A total of 2,280 unique ASVs were detected across the 200 samples, including 2,241 bacteria and 39 archaea (Table S5). The majority of ASVs were distributed across the bacterial phyla Proteobacteria ($n = 1,052$), Bacteriodetes ($n = 325$), Verrucomicrobia ($n = 60$), Marinimicrobia ($n = 54$), and Cyanobacteria ($n = 39$). A total of 350 ASVs belonged to chloroplasts. A large portion (22.3%, $n = 509$) of ASVs were detected only in the coastal environment, compared to 7.4% unique to the offshore ($n = 169$) and 2.8% unique to transition ($n = 63$) stations.

Community composition was further assessed by examining the top 14 most abundant orders measured as a proportion of the entire community. Synechococcales was the

most abundant order detected in the coastal environment (26.3 ± 10.6%; mean ± s.d., $n = 120$), while Flavobacteriales (24.7 ± 8.1%, $n = 40$) and Pelagibacterales (31.6 ± 7.6%; $n = 40$) were the most abundant orders detected in the transition and offshore environments, respectively (Fig. 2A). A total of eight of the 14 orders examined showed significant differences between all comparisons of environment types (coastal *vs.* transition, transition *vs.* offshore and coastal *vs.* offshore, $p < 0.001$) when evaluated with DESeq2 normalization (Pelagibacterales, Rhodobacterales, Puniceispirillales, SAR86, Actinomarinales, Betaproteobacteriales, Euryarchaeota Marine Group II, and Parvibaculales; Table S5). The remaining orders showed significant differences across two comparisons of environment types (coastal *vs.* transition and transition *vs.* offshore: Synechococcales, Rickettsiales, Flavobacteriales; coastal *vs.* offshore and transition *vs.* offshore: chloroplast and Cellvibrionales; coastal *vs.* transition and coastal *vs.* offshore: Oceanospirillales; $p < 0.001$, Table S6).

Within the Synechococcales, the combination of *Synechococcus* and *Prochlorococcus* compromised between 89.0 ± 4.4% and 98.0 ± 2.5% of the total Synechococcales relative abundance at a given station (Fig. 2B). *Synechococcus* dominated the microbial community in the coastal stations (27.4 ± 10.5%, mean ± s.d., $n = 120$), declining in the transition (12.5 ± 8.3%, $n = 40$) and offshore environments (10.7 ± 8.0%, $n = 40$) (Fig. 2B). In contrast, *Prochlorococcus* was most abundant in the offshore environment (13.3 ± 0.1%), declining in the transition (3.3 ± 0.0%) and coastal environments (0.3 ± 0.0%). The relative abundance of *Synechococcus* was roughly two-fold higher in sequence data than in flow cytometry data (Figs. 2B–2C). In contrast, *Prochlorococcus* had similar relative abundances between sequence data and flow cytometry data.

The seasonality of phytoplankton taxa was evaluated using DESeq2 normalized comparisons of chloroplast, *Synechococcus*, and *Prochlorococcus* total abundance as defined by sequence data. No significant seasonal patterns in chloroplast abundance were detected across the three environments (Wald test, $p > 0.05$ all comparisons, Fig. S4). In the coastal environment, *Synechoccocus* increased in spring and summer (winter *vs.* spring, $p < 0.001$; summer *vs.* winter, $p = 0.015$; fall *vs.* spring, $p = 0.042$). *Synechococcus* decreased in the winter and increased in the summer in both the transition (winter *vs.* summer, $p = 0.049$; winter *vs.* spring = 0.009) and offshore environments (winter *vs.* summer, $p = 0.002$; winter *vs.* spring, $p = 0.014$). *Prochlorococcus* showed an increase in fall in the coastal environment (fall *vs.* spring, $p = 0.006$; fall *vs.* summer, $p < 0.001$; winter *vs.* summer, $p < 0.001$ Fig. S4).

## Spatial and temporal distributions of SAR11 subclades

SAR11 accounted for 106 ASVs distributed across eight subclades: Ia, Ib, IIa, IIb, IIIa, IV, Va, and Vb (Table S7). Individual samples harbored between 7 and 55 SAR11 ASVs, averaging 14 ± 4 SAR11 ASVs in the coastal environment (mean ± s.d., $n = 120$), 17 ± 6 in the transition environment ($n = 40$), and 28 ± 9 in the offshore environment ($n = 40$). Generalized linear models with Poisson distributions revealed that the number of SAR11 ASVs recovered per sample varied significantly across environments (Likelihood Ratio Test (LRT) = 1168, $p = 0.001$), but did not vary significantly across seasons

(LRT = 73.49, $p$ = 0.114). While sequencing depth did not vary over environment or station (Table S4), we do note that significant seasonal differences in sampling depth could obscure these results (Table S3). Of the eight SAR11 subclades identified across KByT, subclade IIa harbored the highest number of unique ASVs ($n$ = 41), followed by Ia ($n$ = 22) and Ib ($n$ = 19) (Table S7). Within individual subclades, the average number of ASVs detected in a sample significantly differed between environments for subclades Ib (LRT = 641.52, $p$ = 0.001), IIa (LRT = 194.92, $p$ = 0.001), IIb (LRT = 19.31, $p$ = 0.001), and Vb (LRT = 13.78, $p$ = 0.001) (Table S7).

Collectively, the relative abundance of SAR11 increased from coastal stations (21.6 ± 4.7%, mean ± s.d., $n$ = 120), to transition (23.2 ± 6.4%, $n$ = 40) and offshore stations (34.2 ± 7.2%, $n$ = 40). DESeq2 normalization revealed significant differences in total SAR11 abundance between the three environments (Wald test, coastal *vs.* transition, $p < 0.001$; transition *vs.* offshore, $p < 0.001$; coastal *vs.* offshore, $p < 0.001$, Fig. S5). As a whole, the abundance of SAR11 did not significantly differ across seasons within the transition or offshore environments, but did in the coastal environment where it reached its maximum in spring (winter and spring $p$ = 0.002; fall and spring, $p < 0.001$; winter and summer, $p$ = 0.050; fall and summer, $p$ = 0.035; Fig. S5).

SAR11 subclade Ia was the most abundant subclade detected across all stations, with a relative abundance that was fairly consistent across coastal (14.6 ± 3.9%, mean ± s.d., $n$ = 120), transition (14.2 ± 3.9%, $n$ = 40), and offshore (14.3 ± 2.3%, $n$ = 40) environments (Fig. 3). In contrast, subclade IIb was the least abundant of all SAR11 subclades detected throughout KByT, reaching a maximum relative abundance of 0.01 ± 0.04% (mean ± s.d., $n$ = 20) at offshore station NTO1. Both subclades Ib and IIa increased in relative abundance in offshore stations compared to their coastal counterparts (Fig. 3). While spatial differences were most pronounced for SAR11 subclades Ib and IIa, significant differences in relative abundance across the coastal to offshore transect were detected for seven of the eight subclades recovered in this study (Fig. S7). DESeq2 normalization revealed that subclades Ib, IIa, IV, and Vb were more abundant in offshore compared to coastal stations (Wald test; $p < 0.001$), while subclades Ia and IIIa were more abundant at coastal stations compared to the offshore ($p < 0.001$). Subclades Ib, IIa, IV, and Va also differed between the coastal and transition environments ($p < 0.001$, except for IV where $p$ = 0.002), while subclades Ia, Ib, IIa, IIIa, IV, Va, and Vb differed between transition and offshore environments ($p < 0.001$, except for Ib where $p$ = 0.017).

Significant seasonal differences in the abundance of individual subclades were observed in both coastal and offshore environments, but not in the transition environment (Fig. S8). These include (i) subclade IIIa, which decreased during winter in both the coastal (Wald test, winter *vs.* fall, $p$ = 0.003; winter *vs.* spring and winter *vs.* summer, $p < 0.001$) and offshore environments (winter *vs.* fall, $p < 0.001$; winter *vs.* spring, $p$ = 0.002; winter *vs.* summer, $p$ = 0.005); (ii) subclade Va, which increased in spring in the coastal environment (winter *vs.* spring, $p < 0.001$; spring *vs.* fall, $p$ = 0.029); (iii) subclade Ia, which increased during winter in the coastal environment (winter *vs.* fall, $p$ = 0.002; spring *vs.* fall, $p$ = 0.041); (iv) subclade IV, which increased in spring in the coastal environment (winter *vs.* spring and spring *vs.* summer, $p < 0.001$); and (v) subclade Vb,

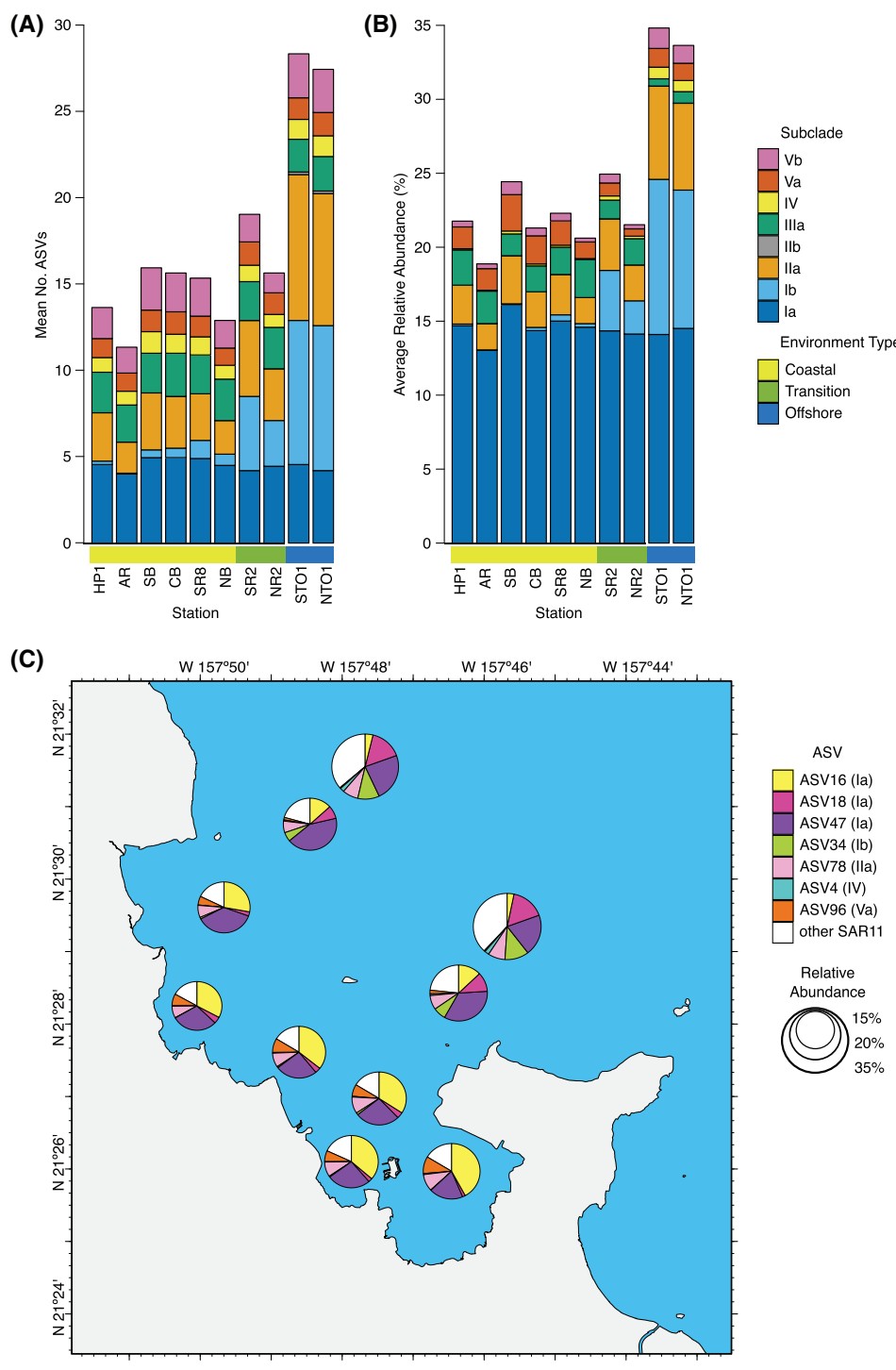

**Figure 3 Spatial distribution of SAR11 subclades.** (A) Average number of SAR11 ASVs per subclade, and (B) subclade relative abundance across KByT stations. n = 20 samples per station. (C) Average proportion of the total SAR11 relative abundance for the ubiquitous SAR11 AS*Vs*. Seven ASVs are either ubiquitous within coastal (ASV16, ASV96), offshore (ASV34, ASV18, ASV4), or both (ASV47, ASV78) environments. Circle size indicates the relative proportion of SAR11 in the total microbial community. Map plotted in ArcGIS Pro 2.7.

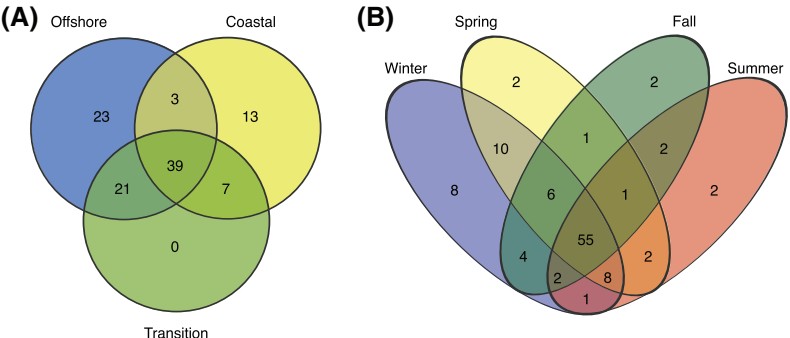

**Figure 4 Environmental and seasonal distributions of SAR11 ASVS.** Venn diagrams comparing the distribution of total number of ASVs detected across (A) environments and (B) seasons.

which decreased during winter in the coastal environment (winter *vs.* summer, $p = 0.018$; winter *vs.* spring, $p = 0.011$).

## Spatial and temporal distributions of SAR11 ASVs

Of 106 SAR11 ASVs in total, 39 were found at least once in each of the coastal, transition, and offshore environments (Fig. 4A). A total of 20 were unique to the coastal/coastal + transition stations and 44 were unique to the offshore/offshore + transition stations (Fig. 4A). All subclades except IIb, which was exceedingly rare overall, contained differentially distributed ASVs across the three environments when evaluated by DESeq2 normalization (Fig. 5; Table S8).

Over half (55) of the 106 SAR11 ASVs appeared across all four seasons (Fig. 4B). The fewest number of SAR11 ASVs were detected in the fall and summer (73 ASVs and $n = 40$ samples each), while a higher number of SAR11 ASVs were detected during spring (85 ASVs in $n = 60$ samples) and winter (94 ASVs in $n = 60$ samples). Some ASVs ($n = 14$) were restricted to a single season: two in each of spring, summer, and fall, and eight in winter. All of the season-specific ASVs also had restricted spatial distributions, were infrequently recovered ($n = 2–5$ samples), and were relatively low in abundance (<1.5% of the SAR11 community). Using DESeq2 normalization, 28 ASVs showed seasonal differences in at least one of the three environments (Fig. 5; Table S8). While six of these seasonal ASVs had significant seasonal differences across two ($n = 5$) or all three environment types (*i.e.*, coastal, transition, offshore; $n = 1$), the majority of seasonal ASVs were detected in the coastal environment ($n = 15$). Three seasonal ASVs were detected in the transition environment and four were recovered in the offshore environment.

## Frequency of SAR11 ASVs across KByT

SAR11 ASVs were grouped into four categories based on the frequency they were detected in samples within each of the three environments: rare (<5%), low-frequency (5–25%); mid-frequency (25–75%); and high-frequency (>75%). In general, most SAR11 ASV diversity was comprised of either rare or low-frequency ASVs, including 45 of 62 ASVs in the coastal environment (73%), 47 of 67 ASVs from the transition environment (70%), and 52 of 86 ASVs from the offshore environment (60%) (Fig. S9). The offshore

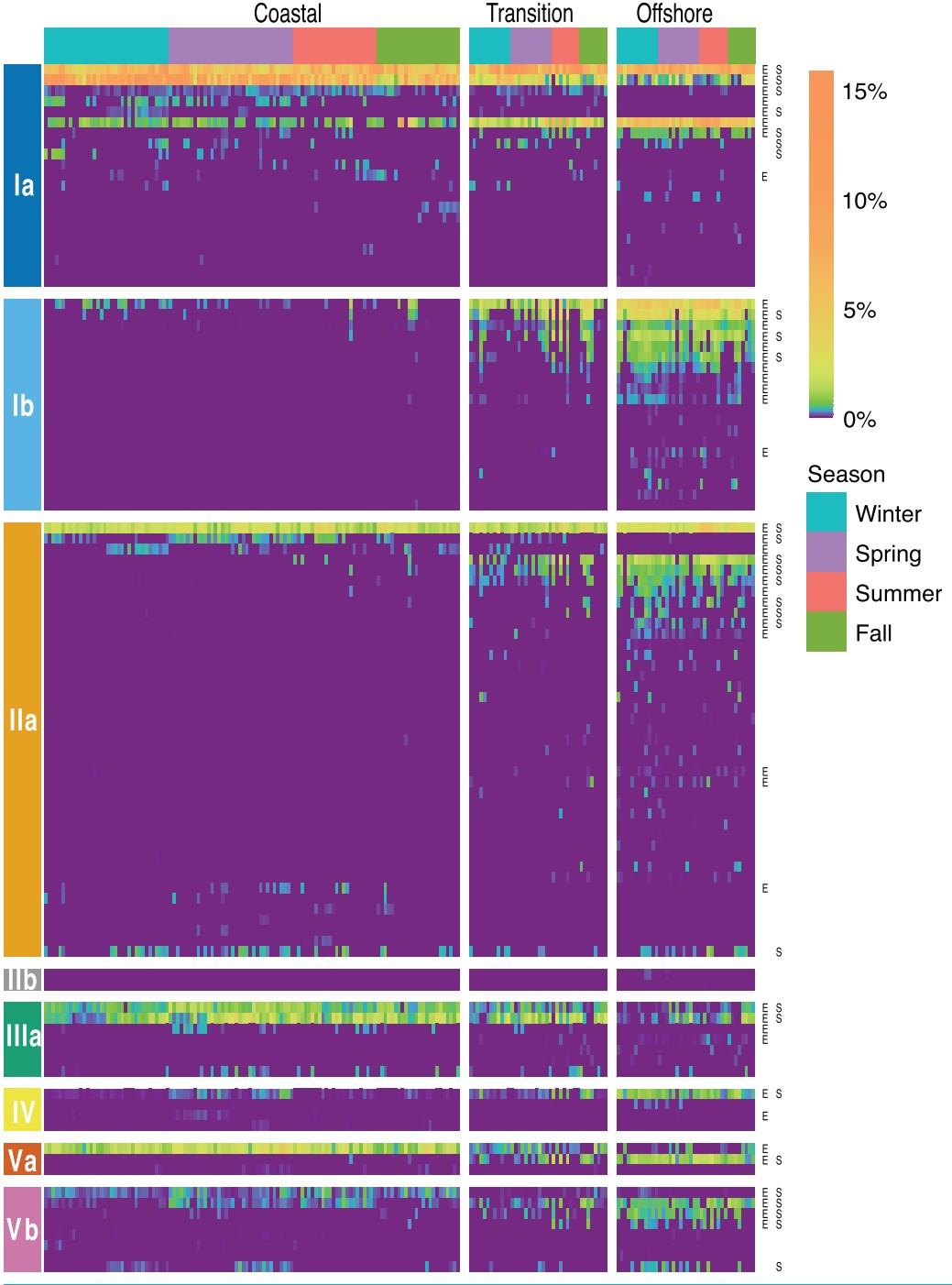

**Figure 5 SAR11 ASV distributions.** Heatmap indicating the relative abundance of SAR11 ASVs (rows) per sample (columns). ASVs are ordered vertically by subclade, and samples are ordered horizontally by environment, season, date sampled, and site. "E" next to ASVs denotes a significant difference between environments, while "S" denotes a significant differences among seasons within a given environment. Wald tests to assess the significance of differences among environment and season for each SAR11 ASV were conducted using DESeq2 normalized counts.

environment harbored 16 ASVs in the high-frequency category, followed by nine in the transition environment and seven in the coastal environment. Of these, only three were high-frequency across all three environments and none were unique to the transition zone.

Rare ASVs were most commonly members of SAR11 subclade IIa (coastal $n = 14$; transition $n = 16$; offshore $n = 8$). All subclades had an ASV that was high-frequency, with the exception of the exceedingly rare subclade IIb. Two high-frequency ASVs were ubiquitous across all 200 samples: ASV47 from subclade Ia and ASV78 from subclade IIa. Two high-frequency ASVs were ubiquitous across all coastal samples: ASV16 from subclade Ia and ASV96 from subclade Va. Three high-frequency ASVs were ubiquitous across all offshore samples: ASV18 from subclade Ia, ASV34 from Ib, and ASV4 from subclade IV.

In general, ubiquitous ASVs were also typically the most abundant. As a proportion of the total SAR11 fraction, ASV47 (28.5 ± 10.9%), ASV16 (24.2 ± 15.9%), ASV78 (8.5 ± 3.0%), ASV18 (6.9 ± 6.9%), ASV34 (3.9 ± 4.8%), and ASV96 (5.1 ± 4.2%) were the most abundant SAR11 ASVs across all samples (mean ± s.d., $n = 200$) (Fig. 3C). One exception was ASV4; at 0.8 ± 0.9% of the SAR11 fraction, it was only the 14th most abundant SAR11 ASV yet ubiquitous in the offshore. The seven ubiquitous ASVs made up 83.0 ± 5.9% (mean ± s.d., $n = 120$) of the total SAR11 community within the coastal environment, 77.9 ± 12.8% within the transition environment ($n = 40$), and 62.8 ± 7.9% within the offshore ($n = 40$) (Fig. 3C).

## Correlations between SAR11, picocyanobacteria, and environmental parameters

Hierarchal clustering of environmental parameters, the abundance of dominant SAR11 ASVs, SAR11 subclades, and *Synechococcus* and *Prochlorococcus* picocyanobacteria, by the magnitude and direction of spearman rank correlations ($r_s$) grouped these measures into two main clusters (Fig. S9). The first cluster contained dominant SAR11 ASVs ASV4, ASV18, ASV34, ASV47, and ASV78, SAR11 subclades Ia, Ib, IIa, IIb, IV, and Vb, *Prochlorococcus*, and environmental parameters (salinity and pH) that increased in the offshore environment, while the second cluster included dominant SAR11 ASVs ASV16 and ASV96, SAR11 subclades IIIa and Va, *Synechococcus*, and environmental parameters including the abundance of heterotrophic bacteria, the concentration of chlorophyll *a*, temperature, and the abundance of eukaryotic picophytoplankton that increased in the coastal environment (Table 1, Fig. S9). Three of seven dominant SAR11 ASVs (ASV78, ASV34, ASV18) had strong positive correlations with the subclades that they belonged to ($r_s > 0.7$) (Fig. S9). Subclade Ib, ASV34 (subclade Ib), and ASV18 (subclade Ia) had strong positive correlations with *Prochlorococcus* sequence abundance ($r_s > 0.7$), while more moderate positive correlations were detected between most other offshore SAR11 subclades and dominant ASVs (*i.e.*, subclades IIa, IV, and Vb; ASV4, ASV78) and *Prochlorococcus* ($r_s = 0.4$–$0.7$). Subclades Ia and IIb and ASV47 showed weak correlations with *Prochlorococcus* ($r_s < 0.4$) as well as weak correlations across most comparisons.

Moderate positive correlations were found between SAR11 subclades IIIa and Va and ASV16 and ASV96 and *Synechococcus* abundance ($r_s = 0.4–0.7$).

## Comparison of KByT SAR11 ASVs and isolated strains

Of the 106 SAR11 ASVs detected in KByT, 11 were 100% identical to cultured strains of SAR11, including seven from subclade Ia (Table S9). This included the most dominant and ubiquitous ASV in this data set, ASV47 within subclade Ia, which was identical to the 16S rRNA gene of fifteen SAR11 strains including several isolated from Kāneʻohe Bay (Table S9). ASV16 and ASV18 from SAR11 subclade Ia each matched 100% to cultivated strains and possessed contrasting patterns of distribution, with ASV16 significantly higher in relative abundance in the coastal environment and ASV18 significantly higher in relative abundance in the transition and offshore environments (Fig. 2B; Table S7). ASV78, the other ubiquitously-distributed ASV and the second-most abundant overall, was a 100% match to strain HIMB58 in subclade IIa. One subclade IIIa ASV (ASV11) with ubiquitous distribution in coastal samples was identical to strain HIMB114. Finally, ASV96 in subclade Va exactly matched strain HIMB59 and was ubiquitously distributed in the coastal environment.

The majority (7 of 11) of ASVs with matches to previously isolated SAR11 strains occurred in high frequency and abundance. The remaining ASVs that matched isolated strains were all from subclade Ia (ASV50, ASV20, ASV22, ASV12), and were less frequent and abundant (Table S9).

## DISCUSSION

The Hawaiian island landmasses provide a useful and convenient platform to investigate the ecotypic differentiation of planktonic marine bacteria over an abrupt environmental gradient. Across 2 years of monthly sampling, the coastal environment within Kāneʻohe Bay, Oʻahu resolved itself as a marine *Synechococcus*-dominated system that contains elevated inorganic nutrients, chlorophyll *a*, and cellular abundances of planktonic marine bacteria. These features are consistent with an "Island Mass Effect", where an increase in phytoplankton biomass proximate to near-island and atoll-reef ecosystems is caused by localized increases in nutrient delivery through physical oceanographic, biological, land-based, and anthropogenic processes (*Doty & Oguri, 1956*). A short, three nautical mile transit from the interior of the bay across a transition zone leads to a *Prochlorococcus*-dominated system characterized by depressed inorganic nutrients, low chlorophyll a concentrations, and lower abundances of planktonic marine bacteria-characteristics typical of offshore waters (*Partensky, Blanchot & Vaulot, 1999*).

The persistent differences between coastal and offshore environments across the KByT transect also manifested in distinct distributions of SAR11 marine bacteria. While SAR11 accounted for roughly 20% of the microbial community within the coastal environment of Kāneʻohe Bay, their relative abundance sharply increased to 30–35% in the offshore waters surrounding the bay. In fact, seven of eight SAR11 subclades detected throughout KByT displayed distinct patterns of distribution across coastal Kāneʻohe Bay and the adjacent offshore, the most dramatic of which were an increase in abundance of

subclades Ib and IIa in the offshore. Increases in these two subclades have been associated with oligotrophic ocean gyres (*Morris, Frazar & Carlson, 2012*) and increased *Prochlorococcus* cellular abundances (*Salter et al., 2014*). Our results show that the environmental differences between coastal Kāneʻohe Bay and the neighboring offshore is a strong determinant of SAR11 subclade distribution, providing further support that adaptations to environmental niches may shape the evolution of a majority of SAR11 marine bacteria (*Vergin et al., 2013*).

In previous studies, abiotic parameters including depth, salinity, temperature, and latitude have been identified to co-vary with the distribution of SAR11 subclades (*Field et al., 1997*; *Carlson et al., 2008*; *Brown et al., 2012*; *Vergin et al., 2013*; *Salter et al., 2014*; *Herlemann et al., 2014*; *West et al., 2016*). We observed a fundamental change in the picocyanobacteria that dominate coastal *versus* offshore surface seawater across the KByT system, and hypothesize that this is a major driver determining the distinct distributions of SAR11 subclades and ASVs across KByT. Different SAR11 subclades have shown to preferentially feed on exudates from *Synechococcus* (*Nelson & Carlson, 2012*), and similar patterns may be possible between SAR11 subclades and *Prochlorococcus* (*Becker et al., 2019*). Some evidence of how SAR11 populations may specialize in coastal *versus* oceanic environments comes from analysis of genomes sequenced from isolated SAR11 strains belonging to subclade Ia (*Schwalbach et al., 2010*; *Giovannoni et al., 2019*). For example, genomic analyses of coastal, high-latitude subclade Ia strain HTCC1062 and ocean gyre, low-latitude subclade Ia strain HTCC7211 have shown that the coastal strain is capable of utilizing glucose for chemoheterotrophic growth, while the ocean gyre strain could not (*Schwalbach et al., 2010*). This appears to be a variable metabolic property within SAR11 subclade Ia, and is more commonly detected in productive environments that offer higher concentrations of labile carbon sources (*Schwalbach et al., 2010*). Recent genomic investigations and co-culture studies are expanding our understanding of the diverse phytoplankton byproducts SAR11 cells are able to metabolize, such as volatile organic compounds (*Halsey et al., 2017*; *Moore et al., 2020*) and dimethyl arsenate (*Giovannoni et al., 2019*), emphasizing the importance of co-evolutionary relationships between SAR11 and phytoplankton (*Braakman, Follows & Chisholm, 2017*; *Becker et al., 2019*), and further suggesting that primary productivity plays a significant role in structuring SAR11 genetic and ecological diversity.

Across surface seawater within the three environments of KByT, the relative abundance of SAR11 was dominated by seven ASVs. This observation was similar to that of Ortmann & Santos, who found that ten abundant SAR11 OTUs represented >80% of all of the SAR11 sequences collected from surface seawaters along a transect from coastal Mobile Bay, Alabama, to the offshore Gulf of Mexico (*Ortmann & Santos, 2016*). Across KByT, part of this dominance may arise from the relatively conserved nature of the 16S rRNA gene fragment, which can mask genomic heterogeneity and associated ecological differentiation (*Chase et al., 2018*; *Chevrette et al., 2019*). High similarity of the 16S rRNA gene despite large genomic variation has been previously reported in the SAR11 subclade, with members of subclade Ia sharing 16S rRNA gene identities of 98% or greater occurring within genomes that share average amino acid identities as low as 71%

(*Grote et al., 2012*). In addition, comparisons among intergenic spacer (ITS) and 16S rRNA gene ASVs have shown that SAR11 ITS ASVs provided more and higher correlations between SAR11 and viral communities, suggesting that important strain-specific interactions remain unresolved by the 16S rRNA gene (*Needham, Sachdeva & Fuhrman, 2017*).

Despite its potential limitations, our analysis of 16S rRNA gene ASVs did show ecologically relevant patterns below the subclade-level. Subclade Ia contained three ASVs (ASV16, 18, 47) within the seven most abundant, which is not surprising considering that it is the most abundant SAR11 subclade in surface oceans overall (*Delmont et al., 2019*). Yet, our observation that ASV16 is more abundant nearshore and ASV18 is more abundant offshore, and their segregation into two different clusters based on correlation analyses, provides evidence that their coexistence may derive from functional differences between closely related SAR11 ASVs that result in their differential distribution across end member environments of this study system.

In our study, spring and summer coincide with the dry season in Hawai'i; a period of increased salinity, increased solar irradiance, and diminished dissolved organic nutrients in the coastal environment due in part to decreased nutrient delivery from freshwater streams (*Cox, Ribes & Kinzie, 2006*; *Bryant et al., 2016*). In contrast to offshore KByT stations that underwent little seasonal change, coastal KByT stations within Kāne'ohe Bay experienced notable seasonal changes during spring including increased salinity, *Synechococcus* and heterotrophic bacteria cellular abundance, and an increase in the total relative abundance of SAR11. Based on our results and those of past studies (*Cox, Ribes & Kinzie, 2006*; *Yeo et al., 2013*), it appears that SAR11 and *Synechococcus* within Kāne'ohe Bay tend to increase in abundance during dry periods and decrease seasonally and periodically (*i.e.*, immediately following storm events) with increased rainfall and nutrient concentrations.

Seasonal changes in the diversity and abundance of individual SAR11 subclades have been observed in temperate (*Salter et al., 2014*; *Meziti et al., 2015*), tropical (*Vergin et al., 2013*), freshwater (*Heinrich, Eiler & Bertilsson, 2013*), and subtropical environments (*Chow et al., 2013*). While significant seasonal changes in the number of SAR11 ASVs were not observed in KByT, we did discern strong seasonal patterns in the relative abundance of SAR11 subclades and within individual ASVs. At the offshore stations of KByT, we found that SAR11 subclade IIIa peaked during fall in a similar fashion to observations in surface waters of the Sargasso Sea (*Vergin et al., 2013*). At the coastal stations of KByT, subclades IIIa, IV, and Va peaked in relative abundance during spring, while subclade Ia peaked during the winter, and subclade Vb peaked during spring. Several factors may contribute to differences in subclade seasonality between the coastal and offshore environments. First, different ASVs that comprise these subclades in each environment may display heterogeneous responses to seasonal changes. Second, seasonal fluctuations in the structure and activity of the phytoplankton and cyanobacterial communities that make up the base of the food web that are specific to the coastal or offshore environment may elicit changes in the growth of SAR11 ASVs. In the coastal environment, the average relative abundances of SAR11 and *Synechococcus* reached their

maximum during spring, with highly significant changes in abundance across multiple seasonal comparisons. Similarly, in the offshore environment SAR11 and *Prochlorococcus* both reached their maximum average relative abundances during the summer, however these changes in abundance were less dynamic, with non-significant comparisons across seasons. It is plausible that the abundance of the major consumer of dissolved organic matter, SAR11, is linked to the abundance of dominant photoautotrophs *Synechococcus* and *Prochlorococcus* in the coastal and offshore environments, respectively (*Malmstrom et al., 2010*). We note, however, that it is also plausible that seasonal changes in terrestrial and eukaryotic phytoplankton-derived dissolved organic carbon might differentially impact the structure and abundance of SAR11 assemblages across the nearshore to offshore continuum investigated here.

Relatively little is known about the genetic structure of natural populations of marine microbes in coastal tropical environments, as most studies have come from time-series in temperate systems (*Gilbert et al., 2009*; *Lindh et al., 2015*) and oligotrophic ocean gyres (*Eiler, Hayakawa & Rappé, 2011*; *Vergin et al., 2013*; *Bryant et al., 2016*). Our time-series analyses from tropical coastal Kāneʻohe Bay to its adjacent offshore system show distinct SAR11 communities across environments, that are dominated by a few highly abundant ASVs. This study contributes to a growing knowledge of how coexisting, closely-related populations of marine bacteria are distributed across environmental gradients.

## CONCLUSIONS

SAR11 is the most abundant organism inhabiting seawater of the global ocean. Yet, due to its immense genetic diversity, an understanding of the determinants underlying its spatial and temporal distributions in natural systems remains limited to specific subclades and environments. Here we show sharp changes in the relative distribution of 16S rRNA gene ASVs, subclades, and total relative abundance of SAR11 bacteria across a coastal to offshore transition in the tropical Pacific Ocean. Nearly all of the SAR11 subclades detected throughout the Kāneʻohe Bay Time-series displayed distinct patterns of distribution across coastal, transition, and adjacent offshore environments. While seasonal patterns of distribution for SAR11 subclades occupying offshore stations were few, the seasonality of SAR11 subclades in the coastal environment were unique and more dynamic. Despite their close proximity and a constant exchange of seawater between stations, our findings suggest that environmental selection shapes the stark spatial and seasonal patterns of distribution in SAR11 lineages we observed. We anticipate that a genomic approach will help to elucidate the underlying mechanisms responsible for this ecotypic differentiation.

## ACKNOWLEDGEMENTS

We thank Catherine M. Foley for her generous help with the creation of maps, Evelyn Hoffman, Helen Li, Rachel Ouye, and Hanako Mochimaru for their laboratory and field assistance, Daniel Schar for assistance with fluorometric measurements, Karen Selph for flow cytometry measurements, Jason Jones, Rebecca Weible, and Evan Barba for assistance with sample collection, Jed Fuhrman for providing mock microbial

communities to use in the 16S rRNA gene analyses, and Brian Powell for his advice regarding the delineation of seasons. Any opinions, findings, and conclusions or recommendations expressed in this material are those of the author(s) and do not necessarily reflect the views of the National Science Foundation. This is SOEST contribution xxx and HIMB contribution xxx.

### Funding

This research was supported by funding from the National Science Foundation (No. OCE-1538628) to MSR and the National Oceanic and Atmospheric Administration (NOAA) Margaret A. Davidson Fellowship (No. NA20NOS4200123) to SJT. This material is based upon work supported by the National Science Foundation Graduate Research Fellowship Program under Grant No. 1842402 to SJT. The funders had no role in study design, data collection and analysis, decision to publish, or preparation of the manuscript.

### Grant Disclosures

The following grant information was disclosed by the authors:
National Science Foundation: OCE1538628.
National Oceanic and Atmospheric Administration (NOAA) Margaret A. Davidson Fellowship: NA20NOS4200123.
National Science Foundation Graduate Research Fellowship Program: 1842402.

### Competing Interests

Michael S. Rappé is an Academic Editor for PeerJ.

### Author Contributions

- Sarah J. Tucker conceived and designed the experiments, performed the experiments, analyzed the data, prepared figures and/or tables, authored or reviewed drafts of the paper, and approved the final draft.
- Kelle C. Freel conceived and designed the experiments, performed the experiments, analyzed the data, authored or reviewed drafts of the paper, and approved the final draft.
- Elizabeth A. Monaghan performed the experiments, authored or reviewed drafts of the paper, and approved the final draft.
- Clarisse E. S. Sullivan performed the experiments, authored or reviewed drafts of the paper, and approved the final draft.
- Oscar Ramfelt performed the experiments, authored or reviewed drafts of the paper, and approved the final draft.
- Yoshimi M. Rii conceived and designed the experiments, performed the experiments, authored or reviewed drafts of the paper, and approved the final draft.
- Michael S. Rappé conceived and designed the experiments, performed the experiments, analyzed the data, prepared figures and/or tables, authored or reviewed drafts of the paper, and approved the final draft.

## Data Availability

The data is available at NCBI SRA: PRJNA706753.

## Supplemental Information

Supplemental information for this article can be found online at http://dx.doi.org/10.7717/peerj.12274#supplemental-information.

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
