# Peer review of "Spatial and temporal dynamics of SAR11 marine bacteria across a nearshore to offshore transect in the tropical Pacific Ocean"

_PeerJ, doi:10.7717/peerj.12274_

## Round 0.1 · original submission · Major Revisions

I have received the opinions of three experts. Some issues have been identified that require moderate revision of your manuscript. Please provide a point-by-point answer to all these along with the revised manuscript.

·

Basic reporting

This is a very well written, straight-forward manuscript addressing SAR11 community structure in an interesting environmental gradient off of O'ahu, Hawai'i. The authors use a monthly time series of 16S amplicon sequencing and sophisticated statistical analyses of limited environmental metadata and SAR11 community structure. The data is present in very clear figures and the manuscript contains ample supplementary figures and tables.

I didnt find accession numbers for the raw data, and there needs to be a mention that sequence information of the ASVs is presented in Table S3.

Experimental design

The experimental design is very straight forward and follows a high standard. The analyses offers additional information on SAR11 ecotype structure and its relation to environmental parameters. Methods are described well.

Validity of the findings

The data is discussed in the context of previous findings on ecotype structure in SAR11. All presented data is robust and conclusions are well stated. The data informs our understanding of ecotype diversity and its relation to environmental parameters. Additional data, both abiotic (e.g. nutrient concentrations, DOC characterization) as well as biotic (composition of eukaryotic phytoplankton) would have been helpful, but should not be required.

Additional comments

This is a well written paper and I only have very minor comments.

1) The speculation on marine picophytoplankton shaping SAR11 ecotype structure (lines 490 - 493) should also mention the possibility of terrestrial sources of DOC or eukaryotic primary producers having a strong impact here.

2) Selection of figures and tables for main manuscript:

Figures 4 and 5 are somewhat redundant and one can be moved to the supplement.

The results start with the discussion of the whole microbial community, and I would suggest moving Fig S4 into the main manuscript.

The tables 1 and 2 are not very informative and can be moved to the supplement. Table 1 is redundant with Fig 4.

Table S1 could be moved to the main text.

3) Title: I would drop 'sampled'

Reviewer 2 ·

Basic reporting

The study by Tucker et al. provides a 2-year time-line series analysis of the abundance of SAR11 (Pelagibacterales) in surface waters along a coast to offshore transect from a tropical region. Its unique location and high resolution (almost monthly sampling) make it an interesting comparison to other studies that investigate SAR11 or bacterioplankton in general. It is well written, and the illustrations are very helpful.

There are several points that should be addressed:

A major flaw is that the sequencing data are not made public available (not even for reviewers). The author should submit their data to a public resource and add a section at the end of the methods, where they indicate the resource (e.g., the SRA accession numbers).

DNA was extracted from a size fraction (85-0.1 um, line 123-125) that includes not only free-living bacteria (see comment in Experimental design). Given that the generated data is a relative abundance, the author should provide a complete ASV table and not just a table for SAR11 ASVs. This could be divided in three tables, one for each of the three habitats. Non-SAR11 taxa could be presented at phyla or kingdom level. This way, it would be possible to see how much the relative abundance of SAR11 is influenced by a potential uneven distribution of picophytoplankton (see line 231) or typically particle-attached bacteria between habitats.

SAR11 clade V should be removed from the analyses as it does not belong to the Pelagibacterales (see e.g., Haro-Moreno et al. 2020 EM or Viklund et al. 2013 PLoS One 8: e78858), which is the group that is analyzed here (e.g., line 26, line 43). The authors appear aware of the problem as the current Silva annotation is marine AEGEAN group 169 in the order Rhodospirillales (line 192), but added it to the study. Is there any particular reason for this?

Experimental design

The authors collected the size fraction 85-0.1 um (line 123-125) to analyze the known free-living (line 47) object of interest, SAR11. Typically, eukaryotes and particle-associated bacteria are removed, when interested in free-living bacteria (e.g., the authors of the paper cited by the authors about their primer choice (Parada et al. 2016) used a 1 um pre-filter, other typical choices are 2.7 um or 5 um). The resulting DNA thus will contain also eukaryotic DNA and particle-associated bacteria. Given that 16S rRNA amplicon data are relative abundances (and the chosen primer will also amplify chloroplasts) this will lead to reduced SAR11 fractions even though the absolute numbers might be the same. While the authors remove chloroplasts and mitochondria data from their SAR11 analyses (line 188-190), a potential difference in particle-associated bacteria between habitats is not addressed. Differences in relative abundance could therefore reflect differences in the number of particle-attached bacteria (e.g., the coastal site might have more particles than the offshore sites). The authors could maybe add some information for their choice of pre-filtration.

The author compare ASV counts and assign frequencies on not subsampled data (e.g., test of difference in ASV numbers line 204-207, frequency assignment line 212-216, as well as results line 288-298, line 335-337, 340-346, 353-368). As the authors got large differences in sequencing depth ranging from 9,393 to 100,278 reads per sample (line 262), these comparison and numbers are hard to interpretate as they could be heavily influenced by differences in sequencing depth. I suggest providing a supplementary table with the sequencing depth for each sample or a summary for each habitat and test if there are significant differences between habitats and seasons within a habitat. Subsampling should be done if species diversity is compared to ensure the same kind of “sampling effort”.

In line 474, the authors refer to alpha diversity, do they refer to the comparison of ASV counts? Otherwise, it is not shown which diversity index they used, and no results are shown. This could be clarified. Again, samples should be subsampled/rarified for alpha diversity comparisons (and samples with extreme differences in reads should be removed).

The authors are interested in ecological differentiation (line 38) and individual (!) SAR11 amplicon sequence variants of SAR11 (line 112) and refer to ecotypes (line line 60). It is therefore not clear why the authors choose a 99.6% similarity threshold (1 mismatch over 245 bp) rather than exact sequence variants (no mismatch) - especially, as the authors refer to the need of single nucleotide resolution in the introduction (line 71-74).

Validity of the findings

The authors hypothesize that differential abundance of SAR11 clades is linked to difference in the picocyanobacterial structure (line 425-428). This is certainly a valid hypothesis also in respect of recent findings about Prochlorococcus (e.g., Roth-Rosenberg et al. 2021 Limnology&Oceanography) and the observed a statistical difference in Prochlorococcus and Synechococcus distribution. However, the authors did not test for covariance with different SAR11 clades. Indeed, no analysis of the influence of environmental factors on the relative abundance of SAR11 clades is made. These analyses would certainly add some explanation for the observed differences and allow comparison to previous findings (line 422-424).

The authors describe that clades Ib and IIa have the most dramatic increase in the offshore and state that in other studies these increases have been associated with decreasing salinity and nutrients (line 415-417). The salinity difference is opposite in KByT, the coastal salinity is lower than offshore one (line 229-230). Therefore, the distribution of these two clades is opposite of the expectation if salinity is the driving factor.

The authors state that there is a sharp increase in SAR11 relative abundance toward the offshore waters (Line 410-413). Could this be due to less particle-associated or other bacteria (e.g., the bacterial counts were lower in offshore stations)? So that absolute numbers would be similar between offshore and coastal environment (and thus there would be an independent carrying capacity for SAR11).

Additional comments

The authors should avoid using the term ecotype/ecotopic for the difference between clades and rather use ecological differentiation. Ecotypes as the authors state (line 60-62) is about closely related bacteria. Each clade of SAR11 contains different genera-level groups (see e.g., Haro-Moreno et al. 2020 EM). For within subclade comparisons the term ecotype might be correct.

The authors should state (especially in the discussion) that they analyzed only the surface water (2m, line 119). SAR11 clades have been shown to have depth specific distribution as I am sure the authors are aware of (e.g., line 422). Hence all the conclusions are only valid for the analyzed layer.

Some minor things:

Line 46-47: Morris et al. 2002 and the 25% are old numbers. Metagenomic analyses and use of recent primers indicate that as much as 40% and more of the microbial community can be SAR11 (e.g., Morris et al. 2012 EM, West et al. 2016 Frontiers in Microbiology, Dubinsky et al. 2017 EM).

Line 53: Please add here also the reference from Jimenez-Infante et al. 2017 that is given later. Maybe add of how many clades isolates are now available.

Line 88: Add references for the 16S-ITS studies.

Line 88-89: Move Jimenez-Infante et al. 2017 to references for cultivation of new strains.

Line 87-91: Add efforts from FISH based studies (e.g., Herlemann et al. 2014, or studies on SAR11 from the Western Mediterranean Sea).

Line 94: One steep gradient is salinity (e.g., in the Baltic Sea). Therefore, the Herlemann et al. 2014 study, cited later, should be mentioned in this paragraph, especially since salinity is mentioned in line 99.

Line 195: Add the used outgroup.

Line 227: The authors should add that the difference is small, but significant between transition and offshore (e.g., only 7.1% of the variability between them is explained due to this separation). The same can be seen in the Venn diagram of shared ASVs (Figure 3A), which is quite high between offshore and transition and low between the other comparisons, and the distribution of Prochlorococcus. Also, it should be added that only two of the eight analyzed factors were different, compared to 6 between coastal and transition.

Line 257-258: Add statistical test used.

Line 274-278: How was the chloroplast distribution along the three habitats?

Line 281-284: Add the statistical test and the test values.

Line 284: The authors could add a short paragraph for the order Pelagibacterales relative abundance before going into the different subclades (done in the same way as Pro/Synechococcus). (e.g., move lines 310-313 and 322-324).

Line 313: Give the test statistics.

Line 314-321: This paragraph seems to repeat descriptions made in the paragraph above (lines 299-313). The two paragraphs should be combined. I suggest removal of mentioning specific stations and rather focus on the three habitats.

Line 322-326: Give statistical test and test statistics when using the words “significant differences” or a p-value.

Line 341: Were in fall and summer 73 ASVs? Maybe clarify.

Line 378-394: To me this paragraph is not about biogeography as it is just a comparison to cultures and not for example to datasets like the TARA ocean. Furthermore, as only a small part of the 16S rRNA gene is analyzed, even a 100% match does not say much about the similarity of the strains. In addition, bacteria with 100% identical 16S rRNA gene similarity can have largely different genomes, which might be true for SAR11 as it the 16SrRNA gene might be a recombination hotspot (Haro-Moreno et al. 2020). When using the word “significantly”, please give test and test statistics. Change abundant (line 380 and 387) to dominant, as the data are about relative abundance. In absolute numbers a coastal ASV might be the highest given the higher number of heterotrophic bacteria.

Line 465: The salinity is only increased compared to the coastal stations at other times. It reaches the same level as offshore.

Line 470: Are there nutrient data available for the bay?

Line 484-493: This is a repetition of the hypothesis raised in line 425. This should be combined.

Line 503-522: The conclusion should be shortened. It seems more like a summary of the results and discussion.

Line 531: Replace the xxx.

Figure 2: A) ASV numbers are not comparable as they are from samples with different sampling depths.
B) Which dataset was used? With chloroplast and mitochondria or without?
Please inverse the order of the legend to match the order in the diagram.

Figure 5 could be given as a panel of Figure 2.

Supplementary Tables: Include Supplementary Table S4 in the file with the other supplementary tables. Please provide for each a headline above the respective table (e.g., as if it is in the main text).

Supplementary Table S3: Please indicate significant values (e.g., make them bold or red) to ease reading.

Supplementary Figures: Please add the descriptions for each supplementary figure below the figure in the pdf file.

Supplementary Figure S3: How was the nMDS calculated (e.g., which software was used)? Give the value of k (e.g., k=2 or k=3). To ease reading, remove station names and use different symbols for the three different habitats. Color filling could be done according to season. Different stations could be indicated by different shades of the same color.

Supplementary Figure S4 and S5: Please give the legend in the same order as in the plot (e.g., inverse the order).

Supplementary Figure S6: Legend and Figure differ in A and B.

Supplementary Figure S10: This is a great figure. Consider giving this in the main manuscript.

Reviewer 3 ·

Basic reporting

No comment, see my full review.

Experimental design

No comment, see my full review.

Validity of the findings

No comment, see my full review.

Additional comments

I found the manuscript “Spatial and temporal dynamics of SAR11 marine bacteria
sampled across a nearshore to offshore transect in the tropical Pacific Ocean” very interesting to read. The authors report their results from surveying the microbial community off O’ahu, HI, clearly a region of global relevance, especially the offshore stations. The nearshore sites provide interesting contrasts to these offshore sites, and indeed distributions and ecology, in particular, of SAR11. I appreciated the summary statistics provided for the nearshore, transition, and offshore sites for the different clades of SAR11 and other groups. I also appreciated the evaluation of the microdiversity of SAR11, showing how different ASVs vary between the sites. These strong gradients showed a surprising degree of variability in some groups of SAR11, though many were found cross all sites. In general, I am supportive of the paper (very interesting and valuable dataset!), but I have some comments and suggestions, see below.

The phylogenetic placements and taxonomy of SAR11:
I think more methodological details are needed for the phylogeny (Figure S2).
What sequences from SILVA (version?) were used (seems a low number)?
I’m unclear how ARB was used for the alignment and/or sequence selection.
What program within GAPPA and more details can be provided?
What options for EPA-NG?
What do the circles on the nodes represent (Figure S2)?

Is there any debate on whether or not the clades IV, Va and Vb are actually SAR11? Especially confusing since manuscript states “AEGEAN-169 within the order Rhodospirillales (for SAR11 subgroup V)”. I would appreciate a comment on this. And again at Line 54: These sequences that at 18% different: are we sure they are all “SAR11”. Do the phylogenetic reconstructions and current genome taxonomy support this (e.g., are they all order Pelagibacterales in GTDB)?

In general, I think the authors would perhaps find interesting this paper which describes SAR11 diversity using high throughput sequencing of the ITS gene: https://www.nature.com/articles/ismej201729. In particular, I think it would be insightful to consider the results from 16S presented in the present MS, with the results from the ITS from that paper. The ITS seems to show remarkably more variability (much less dominance by any single ASV), with an average of about ~0.5% for the dominant ITS-based ASVs vs ~10% for 16S-based ASVs (which seems indeed comparable to the present study). This suggests (unsurprisingly) that 16S is masking potentially important variation, and might be considered at, e.g., Line 443 where a discussion about intra-specific and inter-specific variation occurs.

For Line 443, also might consider calling it intra-clade and inter-clade, as specific would suggest these are species.

I find this sentence in the abstract hard to follow: “On average, 77% of the SAR11 community was compromised of a small number of ASVs (7 of 106 in total) spanning subclades Ia, Ib, IIa, IV, and Va, which were ubiquitously distributed across all samples collected from one or both of the end-member environments sampled in this study (coastal or offshore).” I think the confusing part is “ubiquitously distributed across all sampled collected from one or both”

Line 123: With use of YSI for pH etc, these aren’t really oceanographic grade measurements, I would say, right? Maybe can note these measurements were made with YSI in the figure legends

Line 175: At what stage were Illumina adapters, indexes etc added?

Line 181, can authors report how it was determined that the quality was too low to go forward with the two sequences? Of course, now that only one end is used there has to be lower confidence in that single read. Might be nice to cite other work that has used this single-end approach for Illumina amplicons. Also might increase confidence if a sentence could be inserted to say something like, “Analysis of mock communities using this approach demonstrated accurate denoising of sequences despite only using the single end.”

Line 187, in regards to how mitochondria were treated, unclear to me. I assume “community structure” refers to the multivariate statistics/ordinations like Figure S3?

Line 223, “KByT” here for the first time, but not defined?

Line 270, might consider if Synechococcus seems dominant because they probably tend to have 2 rRNA gene copies? How about a correlation to the flow cytometry?

Line 417, is this cited result (Quéméneur et al) opposite for salinity as shown here?

Line 497, wouldn’t consider Chow et al. 2013 oligotrophic ocean gyre. Coastal, subtropical, maybe.

Figure S8, it would be nice to label these y-axes to say “Average relative abundance as proportion of all SAR11”, or something like that, especially if legends will not be with the figures themselves. Note, I think it would be helpful to label the Supplementary Figures with figure identifiers (Figure S1, S2, etc

---

## Round 0.2 · Minor Revisions

As you can there are a few minor issues that need to be considered before I can make my final decision.

Reviewer 2 ·

Basic reporting

The submitted manuscript has been very nicely revised.

Experimental design

no comment

Validity of the findings

no comment

Additional comments

Minor suggestions
(Note line numbers refer to the track changes file.)

Quote from letter of the submitting author:
"As the reviewer points out, we do indicate at the appropriate place in the Materials and Methods that we sampled surface seawater (line 119). We also do not find any occasions where we are misleading regarding this point, or over-extending our observations to other layers of the water column. If we missed something in the text, please do point it out."

I think this should be clarified in two places Line 555-556 and Line 576-577.
Line 555-556: "We observed a fundamental change in the picocyanobacteria that dominate coastal versus offshore environments across the KByT system,...". The change was only observed in the surface water and not the whole KByT system (which in my opinion would include depth).

Line 576-577 “Across the three environments of KByT, the relative abundance of SAR11 was dominated by seven ASVs” Only the surface layer of each environment was analyzed. I am pretty sure that different SAR11 ASVs will dominate deeper layers of these environments (e.g. at a potential DCM or in non-photic parts). So worth adding here “Across surface waters of the …”.

Other small points

Line 59-60 Consider adding a sentence regarding the uncertainty of clade V in the introduction. Maybe something like “clade V (also known as AEGEAN-169 group)”.

Line 194-196 The data of the mock communities are not shown anywhere, right?

Line 250: Replace “false discovery rate” with “corrected p-value” to avoid confusion with FDR concept, which is a different multiple test correction.

Line 338: Please specify “Euryarchaeota Marine Group II” (also in Table S6).

Line 488-493: State which dominant ASVs, which subclades, and which environmental factors are part of cluster 1 and which of cluster 2.

Line 493-495. To me the sentence is misleading as only 3 of 7 dominant ASVs had strong positive correlations. The four mentioned exceptions are the majority. Could the weaker correlation in these four ASVs be because they belong to clades known to be environmental more variable (e.g. Ia)? Any other possible explanation?

Line 591-599: You could add that ASV16 and ASV18 & 47 are in different clusters (Figure S9) and the latter two in two different subclusters further supporting their ecological differences.

Line 506 to 523: I still do not understand the importance of this paragraph for the article as a whole and suggest deleting it. As the authors point out themselves the 16S rRNA gene in SAR11 is highly conserved (see discussion 583-590). Given that the sequence of only one read is used for the comparison (only 245 bp) for me the comparison is almost meaningless (and again not relevant for this article). It is certainly very interesting if someone is looking at a cultured isolate and wants to know its natural abundance, but again a comparison of only 245 bp is hardly sufficient for this in SAR11. I leave the decision whether to keep it or not to the authors.

Table S6: Could you add columns with relative abundances (or the normalized abundance)? Otherwise, it takes time to find out which order is more abundant in the comparison.

Some typos
Line 337: Pelagibacterales
Line 350 two-fold

Other things
Line 609: Good comment

Reviewer 3 ·

Basic reporting

I approve

Experimental design

I approve

Validity of the findings

I approve

Additional comments

The only remaining issue I have is that the SAR11 clade V issue was not resolved in a convincing way. I suggest adding a comment stating that careful, thorough phylogenomics needs to be done to clarify their relationship to the rest of the SAR11. I started to do my own phylogenomics for the group, ran out of time to do a non-fasttree (there are now a lot of SAR11 genomes, thanks to single cells!), but clearly they are _very different_ than the rest, as are a couple of novel groups: o__TMED127, o__GCA-002720895 (SAR11 groups 6 and 7, perhaps?).

Otherwise, the authors addressed all my concerns, thank you!

---

## Round 0.3 · accepted · Accept

Thank you for addressing all of the reviewers' comments and congratulation on the acceptance of your paper.